# Supersymmetry and multicriticality in a ladder of constrained fermions

**Natalia Chepiga[1]★, Jiří Minář[2,3] and Kareljan Schoutens[2,3]**

**1** Kavli Institute of Nanoscience, Delft University of Technology,
Lorentzweg 1, 2628 CJ Delft, the Netherlands
**2** Institute for Theoretical Physics, Institute of Physics, University of Amsterdam,
Science Park 904, 1098 XH Amsterdam, the Netherlands
**3** QuSoft, Science Park 123, 1098 XG Amsterdam, the Netherlands

★ n.chepiga@tudelft.nl

## Abstract

Supersymmetric lattice models of constrained fermions are known to feature exotic phenomena such as superfrustration, with an extensive degeneracy of ground states, the nature of which is however generally unknown. Here we address this issue by considering a superfrustrated model, which we deform from the supersymetric point. By numerically studying its two-parameter phase diagram, we reveal a rich phenomenology. The vicinity of the supersymmetric point features period-4 and period-5 density waves which are connected by a floating phase (incommensurate Luttinger liquid) with smoothly varying density. The supersymmetric point emerges as a multicritical point between these three phases. Inside the period-4 phase we report a valence-bond solid type ground state that persists up to the supersymmetric point. Our numerical data for transitions out of density-wave phases are consistent with the Pokrovsky-Talapov universality class. Furthermore, our analysis unveiled a period-3 phase with a boundary determined by a competition between single and two-particle instabilities accompanied by a doubling of the wavevector of the density profiles along a line in the phase diagram.

# 1 Introduction: superfrustration and multicriticality

Generally speaking, the physics of a quantum many-body system is determined by a competition between various terms, in particular the kinetic and interaction terms, of its Hamiltonian. For lattice models, endowing the kinetic term with additional density dependent constraints leads to so-called kinetically constrained Hamiltonians, for instance, the East [1–3] and PXP [4–7] models. Focusing on bosonic or fermionic models, such constraints turn out to have dramatic consequences for their physics, resulting for instance in topological ordering [8] or non-thermalizing behaviour following a quantum quench [9–13]. Among lattice models with kinetic constraints, *supersymmetric* models of spin-less fermions play a special role due to an enhanced mathematical control offered by the supersymmetry. Despite this fact, these models come with many outstanding questions. In particular, when going beyond one spatial dimension, it is not only the dynamics, but even the nature of the ground states which remains essentially unexplored. In this work, we consider such a model with a simplest geometry beyond a strictly 1D chain - a zig-zag ladder - and study its ground state phase diagram as we now describe.

**Supersymmetric lattice models**  It has long been known that (space-time) supersymmetry in a quantum field theory (QFT) leads to special features in the physics described by such QFT, and to an enhanced mathematical control. An example of the latter is the Witten index [14] for theories with a complex ($\mathcal{N} = 2$) supersymmetry, which guarantees the existence of zero-energy ground states, without the need of diagonalizing the Hamiltonian. The papers [15–17] traced the $\mathcal{N} = 2$ supersymmetry in critical (conformal) or massive QFT's in 1+1 dimensions to an exact supersymmetry in associated 1D lattice models of spin-less fermions. These supersymmetric lattice models, dubbed the $M_k$ models, have a characteristic constraint, allowing at the most $k$ nearest neighbour sites to be occupied.

**Superfrustration**  Among the $M_k$ models, the $M_1$ model is particularly interesting - it features spin-less fermions with nearest neighbour exclusion principle, forbidding the simultaneous occupation of nearest neighbour sites. Consequently, this allows the $M_1$ model to be formulated on any graph $G$ and there exist proposals how to engineer it in 1D systems using Rydberg atoms exploiting the blockade mechanism to implement the kinetic constraint [18–20].

Evaluating the Witten index for such models beyond 1D led to a surprise: many choices of $G$, such as ladders and 2D grids with underlying triangular, hexagonal or kagome lattice geometries, have a Witten index that is extensive in the system size $N$ (number of vertices of $G$) [21–23]. This implies that the number of ground states grows exponentially with $N$, and that the ground state entropy per site is finite. This phenomenon was dubbed superfrustration in [22].

The ground state counting problem for a variety of choices of $G$ has been addressed, with varying degree of success. For the 2D square lattice with toroidal boundary conditions (BC) precise results have been obtained, establishing that, depending on the choice of toroidal BC, the ground state entropy is at the most sub-extensive, scaling with the linear dimension of the system [24]. For the triangular lattice many results were obtained (bounds on ground state fermion densities [25,26], ground states on ladders [27] and finite patches [28]) but the ground state counting problem remains, to the best of our knowledge, still unsolved.

**Multicriticality**  The massive degeneracy of ground states (which typically come with a range of fermion densities) indicates that supersymmetric points are highly singular points in the ground state phase diagrams of these lattice models. To study this phenomenon, we focus on a relatively simple case, which is the $M_1$ model on a zig-zag ladder. This model, while simple enough to allow powerful numerics, does display superfrustration, with ground states coming in a range of fermion densities $1/5 \leq N_f/N \leq 1/4$. Here and in what follows, $N_f$ is the fermion number. In this paper we study how this model is embedded in a phase diagram set by a Hamiltonian $H$, given in eq. (4), with parameters $t$, $U$, $V_3$ and $V_4$, and reducing to the $M_1$ model for $U = -t$, $V_3 = t$, $V_4 = t$. We explore the vicinity of the supersymmetric point, which turns out to be a multicritical point connecting both gapped and floating phases.

**Methods**  Our main tool has been numerical simulations performed with the state-of-the-art density matrix renormalization group (DMRG) algorithm [29–32] operating directly within the constrained Hilbert space [33, 34]. The explicit implementation of the nearest-neighbor blockade on a zig-zag ladder allows us to reach convergence for critical systems with up to $N = 1201$ sites, keeping up to 1500 states (bond dimension of local tensors) and truncating singular values below $10^{-9}$. Further details of the algorithm will be discussed in Appendix A. Additionally, we supplement the numerics by analytic arguments that are possible in special regions of the phase diagram.

Specifically, considering a parameter space specified by the two interaction strenghts $V_3$ and $V_4$, cf. Sec. 2, we probe a phase diagram described in Sec. 3, where we identify a floating phase (Sec. 3.2) and period-3,4 and 5 density wave phases and analyze the transitions out of them (Sec. 3.3-3.5). We also discuss in Sec. 3.3 how the boundaries of the period-3 phase can be estimated from single- and two-particle instabilities. Furthermore, we provide analytical explanation of the observed valence-bond type state in period-4 and of the density profiles in period-5 phases in Sec. 3.4 and Sec. 3.5 respectively. Sec. 3.6 explores the effect of changing the chemical potential $U$. We conclude in Sec. 4.

## 2  The model

We start by recalling the construction of a supersymmetric model for constrained fermions on a lattice. $\mathcal{N} = 2$ supersymmetry is explicitly realized in the following Hamiltonian built out of two fermionic generators $Q^+$ and $Q^-$

$$H^{\text{SUSY}} = \{Q^+, Q^-\}. \tag{1}$$

Supercharges $Q^+$ and $Q^-$ are constructed from constrained fermions, i.e. the fermions with nearest-neighbor blockade on a selected lattice graph:

$$Q^+ = \sum_i P_i c_i^\dagger, \qquad Q^- = \sum_i c_i P_i, \tag{2}$$

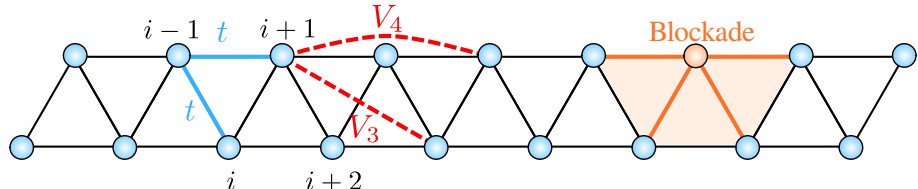

Figure 1: Sketch of the system governed by the Hamiltonian Eq. (4). The hopping $t$ and the interaction terms $V_3, V_4$ are highlighted as solid blue and dashed red lines respectively. The orange region indicates the radius of the blockade where double occupancy is excluded, $n_i(1 - n_i) = n_i n_{i+1} = n_i n_{i+2} = 0$, $\forall i$.

where $P_i = \prod_{<i,j>}(1 - n_j)$ is a projector onto a constrained Hilbert space, with $n_j = c_j^\dagger c_j$ the local occupation operator and $< i, j >$ denoting nearest neighbours on the lattice. On a generic graph the $\mathcal{N} = 2$ supersymmetric Hamiltonian can therefore be rewritten as:

$$H^{\text{SUSY}} = \sum_{<i,j>} (P_i c_i^\dagger c_j P_j + P_j c_j^\dagger c_i P_i) + \sum_i P_i . \tag{3}$$

In this paper we investigate the emergence of the supersymmetric point on a zig-zag ladder. Our many-body Hamiltonian acts on a restricted Hilbert space with $n_i(1 - n_i) = n_i n_{i+1} = n_i n_{i+2} = 0$ and is given by

$$H = t \sum_i (c_i^\dagger c_{i+1} + c_i^\dagger c_{i+2} + \text{h.c.}) + 4U \sum_i n_i + 2V_3 \sum_i n_i n_{i+3} + V_4 \sum_i n_i n_{i+4} . \tag{4}$$

The first two terms describe constrained nearest-neighbor hopping, the third term plays the role of a chemical potential and controls the density, while the last two terms describe the repulsion beyond the blockade. A sketch of the lattice geometry and the Hamiltonian terms can be found in Fig.1. Supersymmetry emerges when $-U/t = V_3/t = V_4/t = 1$, and (4) reduces to (3). Without loss of generality we fix $t = 1$. In addition we fix $U = -1$ and explore the vicinity of the supersymmetric point in the remaining two-dimensional parameter space of coupling constants $V_3$ and $V_4$.

In quasi-1D systems (chains or ladders) with open boundary conditions (OBC) the fermionic nature of the particles does not manifest itself due to local constraints. So fermionic operators in the Hamiltonian eq. (4) can be replaced by hard-boson operators with nearest and next-nearest neighbor blockade. We also note that the Hamiltonian (4) preserves the total number of particles $N_f = \sum_i n_i$.

The supersymmetric point $U = -1$, $V_3 = 1$, $V_4 = 1$ is accompanied with massive degeneracy of zero-energy ground states. For periodic boundary conditions (PBC), the supersymmetric Hamiltonian is $H^{\text{SUSY,PBC}} = H + N$, with $N$ the number of lattice sites. For OBC, supersymmetry requires the following boundary terms

$$H^{\text{SUSY,OBC}} = H + N + 2n_1 + n_2 + n_{N-1} + 2n_N . \tag{5}$$

As a simple example, consider $N = 5$. For PBC, the Witten index is

$$W = \text{Tr}\left[(-1)^{N_f}\right] = 1 - 5 = -4 . \tag{6}$$

For $N_f = 1$ and in the single particle basis $\{|1\rangle, \ldots, |5\rangle\}$, where $|i\rangle$ denotes a fermion at site $i$, the Hamiltonian is given by

$$H^{\text{SUSY,PBC}}_{N=5,N_f=1} = \begin{pmatrix} 1 & 1 & 1 & 1 & 1 \\ 1 & 1 & 1 & 1 & 1 \\ 1 & 1 & 1 & 1 & 1 \\ 1 & 1 & 1 & 1 & 1 \\ 1 & 1 & 1 & 1 & 1 \end{pmatrix} . \tag{7}$$

The translationally invariant state $\frac{1}{\sqrt{5}}(1,1,1,1,1)$, with energy $E = 5$, is the superpartner of the state with zero particles. The supersymmetric ground states can be chosen to be the eigenstates of the translation operator $T$ with eigenvalues $t_l = e^{2\pi i l/5}$ with $l = 1, \ldots, 4$.

For $N = 5$, OBC, the Witten index is

$$W = \text{Tr}\left[(-1)^{N_f}\right] = 1 - 5 + 3 = -1. \tag{8}$$

We thus expect at least one supersymmetric ground state with an odd $N_f$. As for (7), for $N_f = 1$ and in the single particle basis $\{|1\rangle, \ldots, |5\rangle\}$, the Hamiltonian is given by

$$H_{N=5,N_f=1}^{\text{SUSY,OBC}} = \begin{pmatrix} 3 & 1 & 1 & 0 & 0 \\ 1 & 2 & 1 & 1 & 0 \\ 1 & 1 & 1 & 1 & 1 \\ 0 & 1 & 1 & 2 & 1 \\ 0 & 0 & 1 & 1 & 3 \end{pmatrix}. \tag{9}$$

The coefficients of the unique supersymmetric ground state $\sum_i v_{\text{GS}}^i |i\rangle$ of energy $E = 0$ are $\boldsymbol{v}_{\text{GS}} = (v_{\text{GS}}^1, \ldots, v_{\text{GS}}^5) = \frac{1}{\sqrt{20}}(1,1,-4,1,1)$.

In [35, 36] an exact expression was found for the generating function $P_N(z) = \text{Tr}_{\text{GS}}\left[z^{N_f}\right]$ of the ground state multiplicity for the $M_1$ model (with OBC) on the $N$-site zig-zag ladder. Unlike in the Witten index Eqs. (6),(8), where the trace is evaluated over the whole Hilbert space, here the trace is taken only over the subspace spanned by the ground states. It turns out that the generating function satisfies the recursion $P_N(z) = zP_{N-4} + zP_{N-5}$ and a general formula can be found and reads

$$P_N(z) = \sum_{f \in \mathbb{Z}} \left( \binom{f}{N-4f+2} + \binom{f}{N-4f+1} + \binom{f}{N-4f} \right) z^f. \tag{10}$$

We note that $P_5(z) = z$, i.e. there is a single ground state for $N = 5$ and OBC, in agreement with the explicit example discussed above, cf. Eqs. (8) and (9). The formula (10) reveals a massive (extensive) degeneracy of $E = 0$ ground states at the supersymmetric point, at densities in the interval $1/5 \leq N_f/N \leq 1/4$. Their total number grows like $1.167^N$ [35, 36]. This clearly raises the question of the phase diagram in the vicinity of the supersymmetric point, which should be such that ground states at densities in the given interval all meet at a highly singular point in phase space.

## 2.1 Mapping to spins

Before analyzing the phase diagram we present a reformulation, first explored in [35], of the model (with OBC) in terms of unconstrained spin-$\frac{1}{2}$ degrees of freedom. Allowed configurations of the zig-zag ladder model are sequences of 0 and 1 with each 1 accompanied by at least two 0 on both sides. Now add '0' at ficticious sites $i = 0$ and $i = N + 1$ and then substitute $010 \rightarrow \uparrow$ and then $0 \rightarrow \downarrow$ for the remaining 0. This gives a sequence of $N_\uparrow = N_f$ up spins and $N_\downarrow = N + 2 - 3N_f$ down spins on a chain of length $N_s = N_\uparrow + N_\downarrow = N + 2 - 2N_f$. Note that the number of spin degrees of freedom depends on the density of the fermions. Translating

various terms in $H^{\text{SUSY,OBC}}$ into the spin language gives

$$
\begin{aligned}
H_{\text{spin}}^{\text{SUSY,OBC}} = N_s - 2 &- 2 \sum_{j=1}^{N_s} \frac{1 + \sigma_{j+1}^z}{2} \\
&+ \sum_{j=1}^{N_s-1} (\sigma_j^+ \sigma_{j+1}^- + \sigma_i^- \sigma_{j+1}^+) + \sum_{j=1}^{N_s-2} (\sigma_j^+ \frac{1 - \sigma_{j+1}^z}{2} \sigma_{j+2}^- + \sigma_j^- \frac{1 - \sigma_{j+1}^z}{2} \sigma_{j+2}^+) \\
&+ 2 \sum_{j=1}^{N_s-1} \frac{1 + \sigma_j^z}{2} \frac{1 + \sigma_{j+1}^z}{2} + \sum_{j=1}^{N_s-2} \frac{1 + \sigma_j^z}{2} \frac{1 - \sigma_{j+1}^z}{2} \frac{1 + \sigma_{j+2}^z}{2} \\
&+ 2 \frac{1 + \sigma_1^z}{2} + \frac{1 - \sigma_1^z}{2} \frac{1 + \sigma_2^z}{2} + \frac{1 + \sigma_{N_s-1}^z}{2} \frac{1 - \sigma_N^z}{2} + 2 \frac{1 + \sigma_N^z}{2} \ .
\end{aligned}
\tag{11}
$$

We remark that spin Hamiltonians with tri-linear coupling similar to those in $H_{\text{spin}}^{\text{SUSY,OBC}}$ have been proposed in the context of superconducting circuits [37] and recently realized experimentally [38]. It would be thus interesting to investigate whether one could engineer and quantum simulate the specific Hamiltonian (11) in these systems.

## 3 Phase diagram

### 3.1 Overview

We explore the vicinity of the supersymmetric point by tuning the coupling constants $V_3$ and $V_4$ of the Hamiltonian Eq. (4) (with $t = 1$, $U = -1$). Our main results are summarized in the phase diagram in Fig. 2. It contains three gapped density-wave phases with periodicity three, four and five; and a critical Luttinger liquid phase with the density not frozen to a specific value but continuously tuned by two coupling constants. The latter results into incommensurate quasi-long-range order, known in the literature as a floating phase [39]. The supersymmetric point emerges as a multicritical point between the gapped period-4 and period-5 phases and the critical floating phase with the density range $1/5 < N_f/N < 1/4$.

In the period-3 phase every third site is occupied by a fermion while the hopping is completely suppressed due to local constraints. Using a notation $(n_i, \ldots, n_{i+p})$ for a repeated pattern of period $p$ of particle densities, the resulting three ground states are classical of the form $(1,0,0)$ (and $(0,1,0)$ and $(0,0,1)$ respectively) and are decoupled in the sense they are invariant under the action not only of the full Hamiltonian Eq. (4) but also of its individual terms. They also span (in the thermodynamic limit) the $N_f/N = 1/3$ sector and are thus decoupled from each other and the rest of the Hilbert space. An example of the local density profile in the period-3 phase is presented in Fig. 3(a). In the period-5 phase the density is fixed to $1/5$. Here, every fifth site is occupied, however, by contrast to the period-3 phase, quantum fluctuations are not suppressed. As a result the density profile is different from $(1,0,0,0,0)$, as shown in Fig. 3(c). The density of the period-4 phase is fixed to $1/4$. The density profile is very different from the previous two cases: single particles resonate between two nearest-neighbor sites followed by two empty sites (see Fig. 3(b)). Based on the numerical results, we have phenomenologically established that the valence-bond solid type pattern $(0.5, 0.5, 0, 0)$ is an exact ground state along a line $V_3 = V_4$ for $0.8 \lesssim V_3, V_4 \leq 1$, see the dash-dotted line in Fig. 2.

The remaining part of the phase diagram is occupied by a floating phase. The density of the ground state varies inside the phase and is controlled by both coupling constants $V_3$ and $V_4$. In systems with OBC open edges act as a local impurity and induce Friedel oscillations. In turn, the latter lead to standing density waves, a few examples of which are provided in Fig. 3(d)-(f). Below we provide further details on the phases and phase boundaries.

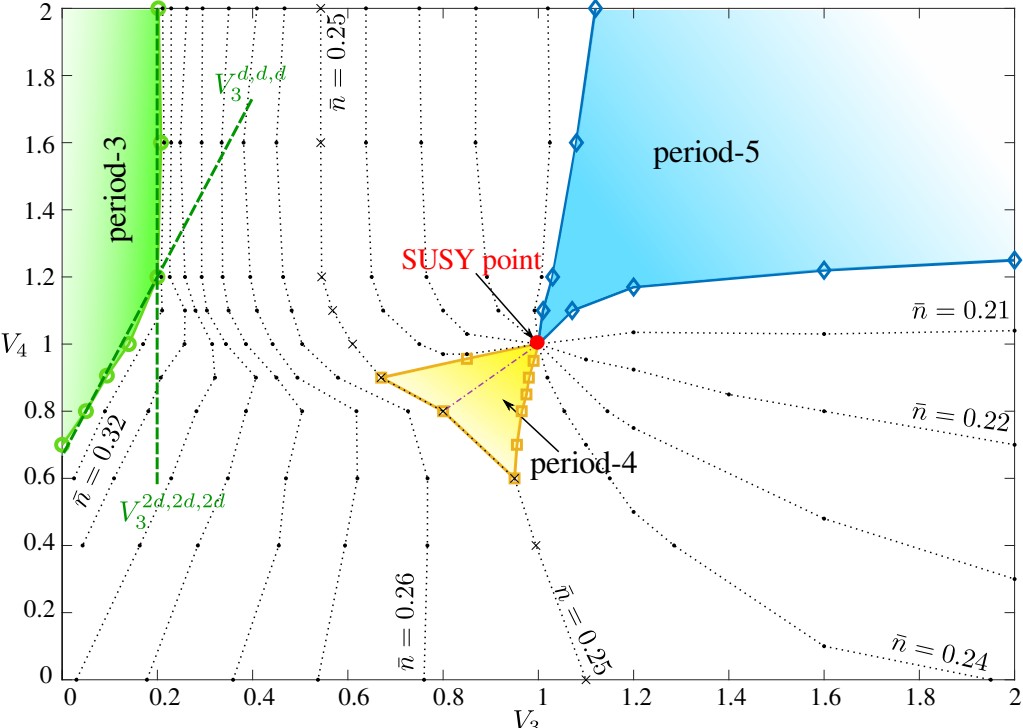

Figure 2: Phase diagram of constrained fermion model Eq. (4) on a zig-zag ladder as a function of third and fourth nearest neighbor interactions $V_3$, $V_4$. The supersymmetric point (red dot) is a multicritical point between period-4 (yellow) and period-5 (blue) gapped phases and the floating phase. At small $V_3$ and $V_4$ sufficiently positive, the system is in the classical period-3 phase. Dotted lines are equal-density lines inside the floating phase. The dash-dotted line marks the location where the valence-bond solid type pattern $(0.5, 0.5, 0, 0)$ is the exact ground state. The dashed green lines indicate the estimated boundaries of period-3 phase delimited by the leading instabilities given by the three single and double defects $V_3^{d,d,d}$ and $V_3^{2d,2d,2d}$, cf. Sec. 3.3.

## 3.2 Floating phase

In the thermodynamic limit the density changes continuously inside the floating phase. We extract the density by averaging the local density over a number of sites as

$$\bar{n} = \overline{\langle n_i \rangle} = \sum_{i=i_{\min}}^{i_{\max}} \frac{n_i}{i_{\max} - i_{\min} + 1} \,. \tag{12}$$

The interval $[i_{\min}, i_{\max}]$ over which we average always lies between two local maxima, as indicated by the red arrows in Fig. 3(d)-(f). This way, even if the wave-vector $q$ is close to a commensurate value (which is in particular relevant in the vicinity of the periodic density-wave phases, cf. also Figs. 4,7,8,11), we obtain meaningful results. To reduce the edge effects we start with the maxima located at a distance of 20-80 sites from the edges. In Fig. 4 we demonstrate how the density changes across the floating phase for two selected cuts across the phase diagram. In Fig. 4(a) the cut at $V_4 = 0.8$ goes through the period-4 phase reflected in a pronounced plateau at $1/4$ filling. Fig. 4(b) shows how the density changes between the period-3 and period-5 phases in the upper part of the phase diagram along $V_4 = 2$. By mapping the density profiles throughout the phase diagram we extract equal-density lines shown in the phase diagram in Fig.2.

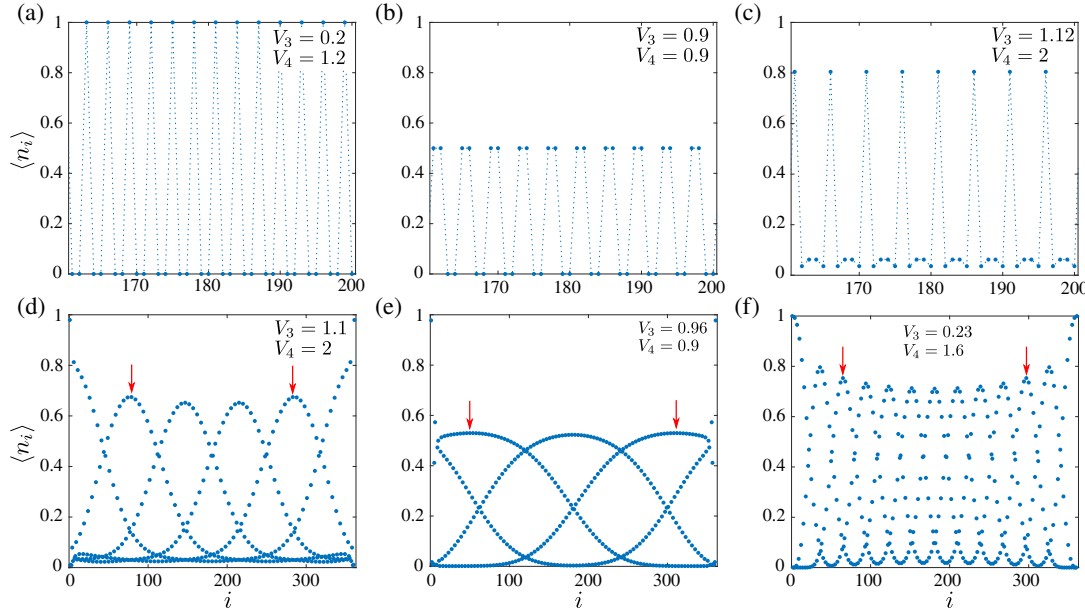

Figure 3: Local density profiles inside (a) period-3, (b) period-4, (c) period-5, and (d)-(f) floating phases. All presented profiles are computed for $N = 361$. For clarity, panels (a)-(c) show only the central part of the profile. In (d)-(f) the average density has been calculated between the peaks marked by the red arrows.

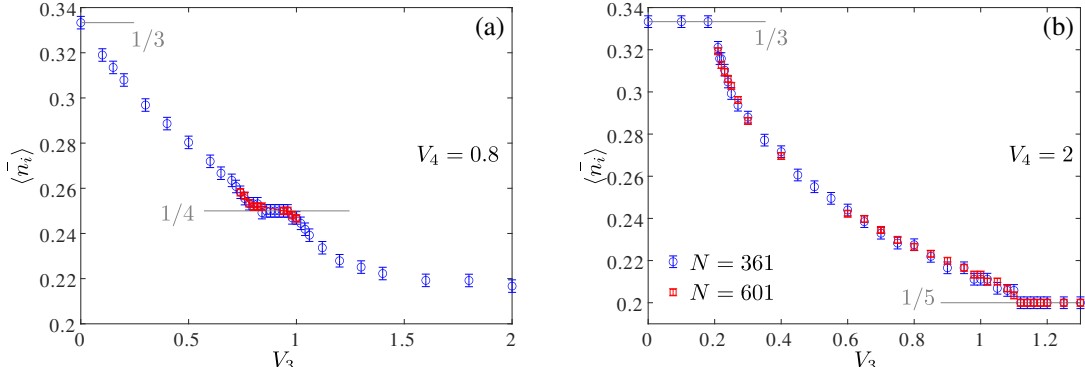

Figure 4: Average density $\bar{n}$ as a function of coupling constant $V_3$ along (a) $V_4 = 0.8$ and (b) $V_4 = 2$. The density changes continuously inside the floating phase. Finite-size corrections are reflected in errorbars and are equal to $1/N$. (a) A plateau at $1/4$ corresponds to the ordered period-4 phase. In (b) the density interpolates between the period-3 and period-5 phases through the floating phase. The blue and red data points correspond to system sizes $N = 361$, $N = 601$ respectively (and analogously in subsequent figures).

Surprisingly, part of the floating phase with the density range $1/5 < \bar{n} < 1/4$ collapses into the supersymmetric point in the "wedge" delimited by the period-4 and period-5 phases, cf. Fig. 2, and re-emerges on the other side of the wedge, within the same density range. The observed collapse leads to an extensive degeneracy of the ground states within the specified density range, resulting in the superfrustration at the supersymmetric point.

The floating phase is a critical phase in the Luttinger liquid universality class. We check this by extracting the central charge from the scaling of the entanglement entropy with the block size in open systems. According to conformal field theory (CFT), the entanglement entropy

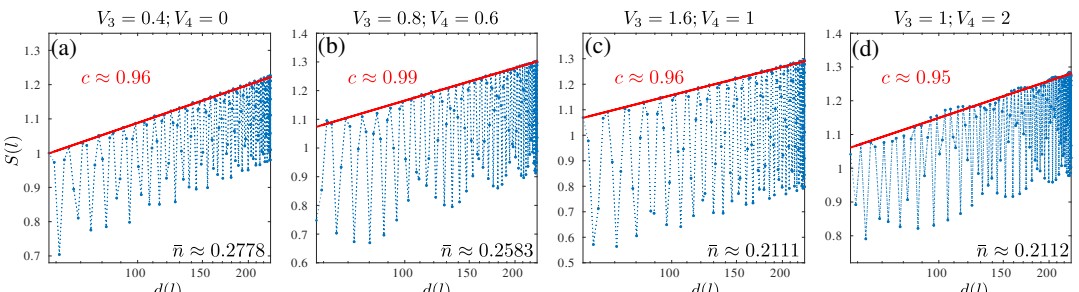

Figure 5: Scaling of entanglement entropy $S(l)$ as a function of conformal distance $d(l) = \frac{2N}{\pi} \sin\left(\frac{\pi l}{N}\right)$ at various points inside the floating phase with $1 \ll l \ll N$ the size of a sub-block of consecutive ladder sites. The numerical values of the central charge extracted from the slope of the entanglement entropy are in good agreement with the theory prediction $c = 1$ for a Luttinger liquid.

scales as [40]

$$S_N(l) = \frac{c}{6} \ln d(l) + s_1 + \log g, \tag{13}$$

where $d(l)$ is the conformal distance $d(l) = \frac{2N}{\pi} \sin\left(\frac{\pi l}{N}\right)$, $l$ is a size of the sub-block $(1 \ll l \ll N)$ and $s_1$ and $g$ are non-universal. Friedel oscillations of the local density cause significant oscillations in the entanglement entropy profile. Nevertheless, when the system size is sufficiently large to accommodate many oscillations they can be safely averaged out, for example, by fitting the local maxima as shown in Fig. 5. The obtained values for the central charge at various points in the floating phase agree within 5% with the CFT prediction $c = 1$ for a Luttinger liquid.

We close this section by commenting on the criticality of ground states at the supersymmetric point. The issue was analysed in [23, 35], by studying the response of the model with PBC to a twist in the boundary conditions. The conclusion was that, while many of the supersymmetric ground states are gapped, some show critical behavior compatible with the number $k$ unitary minimal model of $\mathcal{N} = 2$ superconformal field theory, with central charge $c_k = \frac{3k}{k+2}$, with $k$ even. Our present analysis is complementary to this approach, but it does suggest that, at the supersymmetric point, critical states exist at all (rational) fillings between 1/5 and 1/4. Such states can be obtained by following the floating phase state at given filling into the supersymmetric point.

## 3.3 Boundary of the period-3 phase

An estimate of the location of the boundary of the period-3 phase can be obtained via the following argument. Given the exclusion rule, the only possible density pattern at filling $N_f/N = 1/3$ is of the form ..100100... Let us now imagine taking one particle out $N_f \to N_f - 1$. The resulting 'size 5' hole can split into three independent 'size 3' holes, that is to say, defect patterns $d = ..10010001001..$, where we have highlighted the position of the defect in red. Let us further assume PBC and denote the fully packed state and the state with the defects $d$ as $|\psi\rangle = |..100100..\rangle$ and $|\psi^{d,d,d}\rangle = |..d..d..d..\rangle$, respectively. We then find for the potential energy difference $\Delta V = \langle \psi^{d,d,d}|H(t=0)|\psi^{d,d,d}\rangle - \langle \psi|H(t=0)|\psi\rangle = 4 - 8V_3 + 3V_4$, where $H(t=0)$ is the Hamiltonian Eq. (4) evaluated at zero hopping.

Furthermore, each of the defects $d$ has a unit-strength hopping amplitude, giving a kinetic energy of $\Delta K = -2$ per defect in the large-$N$ limit. Assuming that the defects are far apart and independent gives as energy difference between the defect and the fully packed states

$$\Delta E^{d,d,d} = 3\Delta K + \Delta V = -8V_3 + 3V_4 - 2. \tag{14}$$

Setting $\Delta E^{d,d,d} = 0$ provides the estimated phase boundary, which corresponds to the line $V_3^{d,d,d} = (3V_4 - 2)/8$.

Repeating the exercise for a combination of a 'size 4' hole (double defect $2d$) plus a defect $d$ gives

$$\Delta E^{2d,d} = 4 - 6V_3 + V_4 - 4 = -6V_3 + V_4 , \tag{15}$$

where we used that in leading order each of the holes can hop with unit amplitude. This gives as phase boundary $V_3^{2d,d} = \frac{V_4}{6}$. The relative potential energy for a size-5 hole (triple defect $3d$) is

$$\Delta E^{3d} = 4 - 4V_3 . \tag{16}$$

We note that the triple defect does not involve a direct hopping term, i.e. hopping between two $3d$ defects. More specifically, it is connected by the kinetic term to adjacent pairs of defects of the form $2d, d$ or $d, 2d$. Such terms represent corrections to our simplified treatment, where we only consider defects which are far apart from each other. The relative potential energy $\Delta E^{3d}$ crosses zero at the line $V_3^{3d} = 1$. Taken together, the three lines $V_3^{d,d,d}$, $V_3^{2d,d}$ and $V_3^{3d}$ provide a first approximation to the boundary of the period-3 phase.

Interestingly, we find that for $V_4$ sufficiently positive, the leading instability of the fully packed period-3 phase is to a configuration with $N_f = \frac{N}{3} - 2$ particles. Assuming that the 6 resulting defects group as $2d, 2d, 2d$, we find a potential energy difference $\Delta V = 8 - 10V_3$. The nearest neighbour hopping terms in $H$ of the bare particles, cf. Fig. 1, lead to unit-strength hopping of the double defects $2d$, so that the estimated energy difference is,

$$\Delta E^{2d,2d,2d} = 2 - 10V_3 , \tag{17}$$

putting the phase boundary at $V_3^{2d,2d,2d} = 1/5$.

The instabilities $V_3^{d,d,d}$ and $V_3^{2d,2d,2d}$ meet at the corner $(V_3, V_4) = (1/5,\ 6/5)$, where, in our simple reasoning, they are degenerate with other patterns such as $2d, d$ and $2d, 2d, d, d$. Together these two lines establish an estimated phase boundary of the period-3 phase, which agrees well with the numerical findings, cf. the dashed green lines in Fig. 2.

Clearly, our reasoning here is not exact as (i) there are finite size corrections, (ii) there is a dependence on boundary conditions (open vs periodic) and (iii) the actual lowest energy states at $N_f = \frac{N}{3} - 1$ and $N_f = \frac{N}{3} - 2$ are hybridizations of the single and double defect states described in the above. Nevertheless, the agreement of $V_3^{d,d,d}$ and $V_3^{2d,2d,2d}$ with the numerically found phase boundaries is excellent for sufficiently large system sizes.

In order to get further insight into the nature of the transition out of the period-3 phase we look at the energy level crossings in the immediate vicinity of the phase boundary. Because the total number of fermions is a conserved quantum number, on a finite size system we expect to see a set of explicit level crossings between the fully-packed period-3 state, and the states with one, two etc. particles less. We perform the simulation on systems with OBC and $N = 31$ and $N = 49$. OBC favors first and last sites to be occupied by the fermion, thus the fully packed period-3 state has in total $N_f = \frac{N-1}{3} + 1$ fermions. We look at the level crossing between this state and the states with one and two particles less. Based on the results presented in Fig. 6 we conclude that the single particle instability is a relevant excitation out of period-3 phase for $V_4 \lesssim 1.2$, while for $V_4 \gtrsim 1.2$ the system is driven out of the period-3 phase by the double-fermion instability. In section 3.5 we find that this change of behaviour is connected to a doubling of the dominant density profile wave-vector $q$ in the floating phase. In the vicinity of the period-3 phase, it changes from $q = 2\pi N_f/N$ for $V_4 \lesssim 1.2$ to $q = 4\pi N_f/N$ for $V_4 \gtrsim 1.2$.

The commensurate-incommensurate transition between the ordered and the floating phases is expected to be in the Pokrovsky-Talapov universality class [41]. In Fig. 7 we show how density changes in the vicinity of the period-3 phase boundary. We fit our DMRG data with $A|V_3 - V_3^c|^{\bar{\beta}}$, where $\bar{\beta} = 1/2$ is a Pokrovsky-Talapov critical exponent, and $A$ and $V_3^c$ are

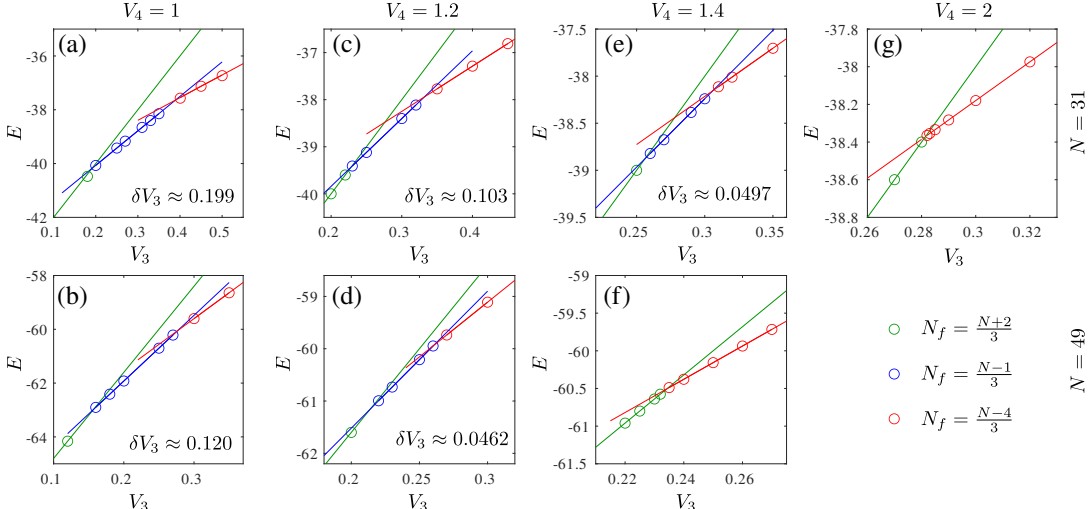

Figure 6: Crossings of the ground state energy for systems with OBC in the vicinity of the period-3 phase boundary and $N = 31$ (top) and $N = 49$ (bottom) between the fully packed state with $N_f = \frac{N-1}{3} + 1$ (green), one fermion less (blue) and two fermions less (red). The interval of the single-particle instability $\delta V_3$ is indicated on panels (a)-(e) and is decreasing with $N$, indicating its finite-size nature. Upon increasing $V_3$ for $V_4 = 1$ (a)-(b) the ground state changes from the fully packed state to a state with one fermion less, then with two fermions less etc. However, starting from $V_4 \approx 1.2$ and above the width of the single-fermion instability vanishes rapidly with the system size and for sufficiently large system the period-3 state changes directly into a state with two fermions less. For $V_4 = 1.4$ we do not observe a single-fermion instability as a ground state for $N = 49$ and larger; for $V_4 = 2$ a single-fermion instability does not show up for systems as small as $N = 31$.

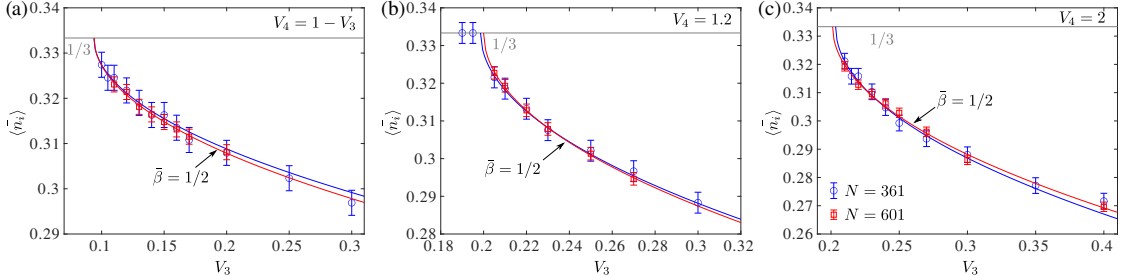

Figure 7: Average density $\bar{n}$ as a function of coupling constant $V_3$ along (a) an oblique cut $V_4 = 1 - V_3$, and two horizontal cuts along (b) $V_4 = 1.2$ and (c) $V_4 = 2$. Solid lines are results of the fit $\propto |V_3 - V_3^c|^{\bar{\beta}}$ with critical value $V_3^c$ as a fitting parameter and Pokrovsky-Talapov critical exponent $\bar{\beta} = 1/2$.

the two fitting parameters. Agreement with the field-theory prediction is remarkable for all cuts, below and above $V_4 = 1.2$, indicating that the double-fermion instability discussed above does not change the nature of the transition in the thermodynamic limit.

## 3.4 Period-4 phase

By contrast to the period-3 and period-5 phases, where (up to quantum fluctuations) a fermion occupies a single site followed then by 2 or 4 empty sites, the density profile of the period-4 phase is fundamentally different: the fermion resonates between two nearest-neighbor sites

followed by two nearly empty sites. This state is reminiscent of a valence-bond solid (VBS) state in quantum magnets.

Phenomenologically we have established that the pattern $(0.5, 0.5, 0, 0)$ corresponds to an (approximate) ground state along a line $V_3 = V_4$ starting from $V_3, V_4 \approx 0.8$ and up to the supersymmetric point $V_3, V_4 = 1$, cf. the dash-dotted line in Fig. 2. At the supersymmetric point, it is easily checked that state (assuming $N = 4N_f - 2$ and OBC)

$$\frac{1}{2^{N_f/2}}(c_1^\dagger - c_2^\dagger)(c_5^\dagger - c_6^\dagger)\dots(c_{N-1}^\dagger - c_N^\dagger)|0\rangle \tag{18}$$

is annihilated by both $Q$ and $Q^\dagger$ and is thus a $E = 0$ supersymmetric ground state of $H^{\text{SUSY,OBC}}$ [35]. We have observed that within the period-4 phase, essentially the same VBS state survives as ground state of $H$ along the line $V_3 = V_4 = a$, $0.8 \lesssim a < 1$.

This is easiest established for periodic boundary conditions (PBC) and $N = 4N_f$, where the VBS state

$$\frac{1}{2^{N_f/2}}(c_1^\dagger - c_2^\dagger)(c_5^\dagger - c_6^\dagger)\dots(c_{N-3}^\dagger - c_{N-2}^\dagger)|0\rangle \tag{19}$$

(together with three similar states obtained by translating (19) over 1, 2 or 3 sites) is an exact eigenstate of $H$ with energy $E = -N - N_f(a-1)$.

For OBC the situation is more subtle. Away from the edges the VBS pattern $(0.5, 0.5, 0, 0)$ is again favored by the Hamiltonian with $V_3 = V_4 = a$, $0.8 \lesssim a < 1$, but there are corrections near the edges. In addition, for $N$ odd ($N = 4N_f \pm 1$), the VBS pattern requires a breaking of the left-right reflection symmetry $i \leftrightarrow N - i$ of the ground state (we indeed observe such symmetry breaking produced in our numerical DMRG procedure). For $N$ even ($N = 4N_f - \sigma$, $\sigma = 0, 2$), the finite size corrections to the PBC ground state can be understood in two ways: either one has to deform the Hamiltonian $H$ at the boundaries to make the VBS state [such as (18)] an eigenstate or one has to modify the state itself to be an eigenstate of $H$. Assuming $N = 4N_f - 2$ for specificity, in the former case if we modify the Hamiltonian into $H^a$, defined as

$$H^a = H + a(2n_1 + n_2 + n_{N-1} + 2n_N), \tag{20}$$

the VBS state eq. (18) is once again an exact eigenstate, with $E = -N + (N_f + 2)(a-1)$, and it is a ground state for $a < 1$ close enough to $a = 1$. In the latter case, considering $H$ instead, the VBS state is modified near the edges. For example, for $N = 10$, $N_f = 3$, and $V_3 = V_4 = 0.9$, the ground state has densities

$$(n_1, n_2, \dots, n_N) =$$
$$(0.7323, 0.2605, 0.0072, 0.0007, 0.4993, 0.4993, 0.0007, 0.0072, 0.2605, 0.7323) \tag{21}$$

recovering thus the pattern $\approx (0.5, 0.5, 0, 0)$ in the bulk.

As shown in Fig. 8(b) the exact line terminates at $V_3 = V_4 \approx 0.8$ with the Pokrovsky-Talapov transition into the floating phase with density $N_f/N > 1/4$. The density scales towards the transition with the Pokrovsky-Talapov critical exponent $\bar{\beta} = 1/2$. The location of the critical point is affected by noticeable finite-size effects. The transition across the remaining two sides of the period-4 phase is into the floating phase with lower density $N_f/N < 1/4$. In Fig.8(a) we show how the density scales approaching the period-4 phase from above (larger $V_4$). The results suggest that the transition remains in the Pokrovsky-Talapov universality class.

### 3.5 Period-5 phase

The density profile in the period-5 phase has a characteristic pattern with peaks every 5 sites and small but non-zero values in between. Below we argue that the change in the local density

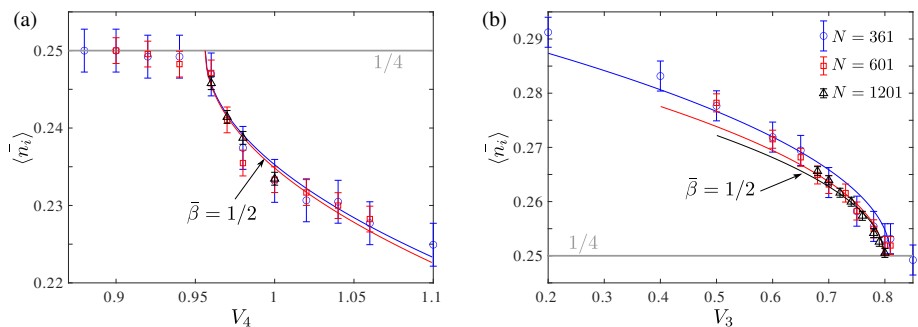

Figure 8: Average density $\bar{n}$ as a function of coupling constant $V_{3,4}$ (a) along the vertical cut at $V_3 = 0.85$ and (b) along the oblique cut $V_4 = V_3$. Solid lines are results of the fit $\propto |V_3 - V_3^c|^{\bar{\beta}}$ with critical value $V_3^c$ as a fitting parameter and Pokrovsky-Talapov critical exponent $\bar{\beta} = 1/2$.

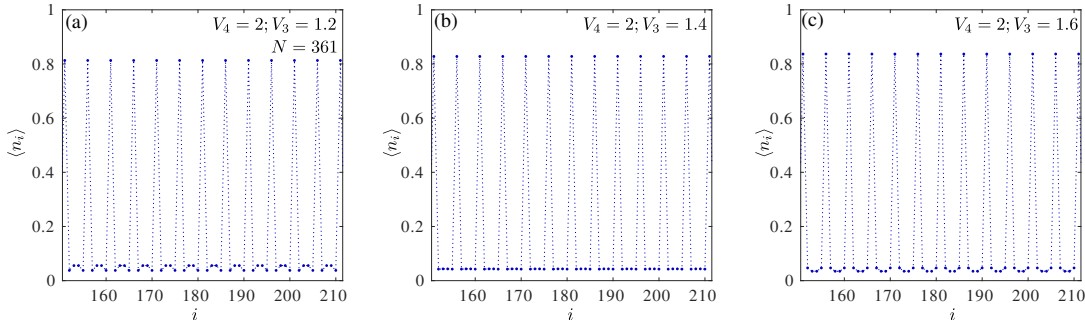

Figure 9: Local density profile for three different points inside the period-5 phase. Change of the dominant wave-vector from $q = 2\pi/5$ ($V_4 = 2$, $V_3 = 1.6$) to $q = 4\pi/5$ ($V_4 = 2$, $V_3 = 1.2$) is reflected with the pronounced change of curvature of the local density between the main period-5 peaks.

curve of the four quasi-unoccupied sites from concave to convex (see Fig. 9) is due to the dominant density wave-vector $q$ changing its value from $2\pi/5$ to $4\pi/5$. The observed doubling of the wave-vector can be understood from the following analysis of a finite size system with PBC, where we add a symmetry breaking term that induces the density profile.

As an example, consider the ladder with PBC, 15 sites, 3 particles. Starting at the supersymmetric point $V_3 = 1$, $V_4 = 1$, we observe 7 degenerate $E = 0$ ground states, with eigenvalues $t_l$ of the translation operator given by

$$t_l = \exp(l\frac{2\pi i}{15}), \quad l = 0, \pm3, \pm5, \pm6. \tag{22}$$

In each of these states the density profile is flat at value $1/5$. We then break the symmetry (both the supersymmetry and the translational invariance) by adding a term

$$H_\epsilon = -\epsilon(n_1 + n_6 + n_{11}). \tag{23}$$

Due to the strict ground state degeneracy, any $0 < \epsilon \ll 1$ immediately breaks the symmetry and leads to a ground state with characteristic period-5 density profile. This same pattern arises in the setting with OBC for large enough system sizes.

The qualitative form of the fluctuations in the density profile after breaking the symmetry can be understood as follows. The perturbation away from the supersymmetric point breaks the ground state degeneracy. For example, it turns out that $V_3 = 1.2$, $V_4 = 2$ gives a pair of ground states with $l = \pm3$, while $V_3 = 1.6$, $V_4 = 1.6$ has two ground states with $l = \pm6$.

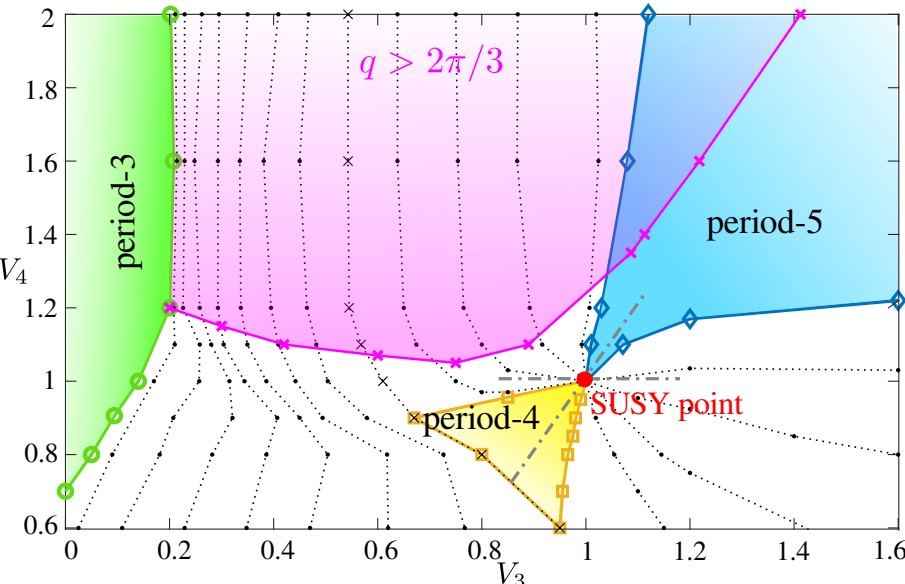

Figure 10: Phase diagram of Fig. 2 with indicated region (magenta), where the dominant ground state density wave vector $q > 2\pi/3$. Dash-dotted lines indicate the two cuts along which we vary the on-site potential $U$ as presented in Fig.12

Turning on $\epsilon$ leads to a ground state that is a superposition of states with all five momenta satisfying $t_l^5 = 1$, but we observe from the numerical solution that the convexity of the density profile follows the pattern at $\epsilon = 0$.

For $V_3 = 1.2$, $V_4 = 2$, among the states with $l = \pm 3$, the symmetry breaking term favors the following combination of these states

$$|v_3\rangle = \sqrt{\frac{2}{5}} \sum_{i=1}^{4} \cos(i\frac{2\pi}{5})|i, i+5, i+10\rangle, \tag{24}$$

which also provides the leading contribution to the density profile

$$n_i = \langle v_3|c_i^\dagger c_i|v_3\rangle = \frac{2}{5}\cos(i\frac{2\pi}{5})^2 = \frac{1}{5}[1 + \cos(i\frac{4\pi}{5})], \tag{25}$$

with $q$-vector of $4\pi/5$. For $V_3 = 1.6$, $V_4 = 1.6$, the analogous analysis with $l = \pm 6$ leads to a density pattern with $q$-vector $8\pi/5$, which is equivalent to $2\pi/5$. This explains the change of the behaviour of the local profiles between the dominant density peaks observed in Fig. 9 from concave to convex.

Scanning the density profiles in the period-5 phase, we find that the $q$-vector is always $2\pi/5$ or $4\pi/5$ depending on $V_3$ and $V_4$. This pattern extends to the rest of the phase diagram, cf. Fig. 10: for low enough values of $V_4$ the $q$-vector equals $2\pi n$ with $n = N_f/N$ the fermion density, but for higher values of $V_4$ we find double that value, $q = 4\pi n$. Interestingly, the line separating the two behaviours connects to the period-3 phase at the cusp point ($V_3 = 1/5, V_4 = 6/5$) where the leading instability out of the period-3 phase changes from 1-particle to a 2-particle instability, as discussed in section 3.3.

In analogy with the commensurate-incommensurate transitions out of the period-3 and period-4 phases, the critical line between the floating phase and the period-5 phase is also expected to be in the Pokrovsky-Talapov universality class. Fig. 11 depicts how the density in the floating phase scales towards the transition. For $V_4 = 1.6$ we show in Fig.11(a) that the density approaches its commensurate value 1/5 with critical exponent consistent with the

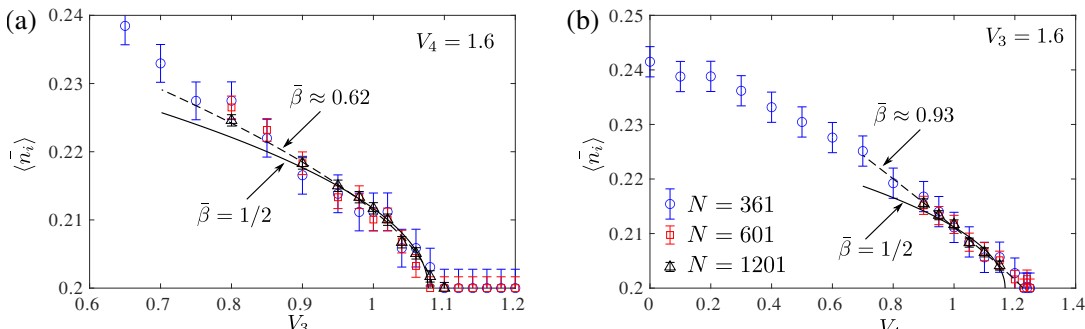

Figure 11: Scaling of the average density $\bar{n}$ inside the floating phase upon approaching the period-5 phase (a) along the horizontal cut at $V_4 = 1.6$, and (b) along the vertical cut at $V_3 = 1.6$. Solid (dashed) lines are results of the fit $\propto |V_{3,4} - V_{3,4}^c|^{\bar{\beta}}$ with fixed (fitted) value of the critical exponent $\bar{\beta}$.

field-theory prediction $\bar{\beta} = 1/2$. We extract an "effective" critical exponent by including $\bar{\beta}$ in the set of fitting parameters. The obtained value $\bar{\beta} \approx 0.62$ agrees within 25% with the Pokrovsky-Talapov value. For $V_3 = 1.6$ presented in Fig.11(b) the computed critical exponent $\bar{\beta} \approx 0.93$ differs significantly from the theoretical expectation. Likely, this is because the presented data points are still too far from the transition to resolve the correct critical behavior. This is supported by the fact that the density in the immediate vicinity of the transition shows fast decrease down to 1/5 with an increasing system size, consistent with the shift of the critical value $V_4^c$ towards the smaller values and followed by the decrease of an effective critical exponent $\bar{\beta}$. Unfortunately, accurate resolution of the Pokrovsky-Talapov transition in these two cases would require an access to much larger system sizes inside the floating phase which is currently beyond the limitation of our computing capacity.

### 3.6 Changing the chemical potential

In order to make the problem tractable, so far we have been focusing at a particular cut of the phase space of the Hamiltonian (4) at a chemical potential $U = -1$. It is thus interesting to investigate the ground state properties where also the chemical potential is varied. To this end Fig. 12 shows the situation along two cuts in the $V_3 - V_4$ plane, namely for $V_4 = 1$ (Fig. 12a) and $V_4 = 2V_3 - 1$ (Fig. 12b). Both cuts are also indicated as dash-dotted lines in Fig. 10, where the former crosses the supersymmetric point from and to a floating phase, while the latter from a crystalline period-4 to a period-5 phase. It is apparent from the results in Fig. 12 that the multicritical nature of the supersymmetric point remains preserved in other planes in the parameter space, namely that the period-4 and period-5 phases remain always separated by a floating phase with varying particle density.

## 4 Conclusion

The model studied in the paper is a prototypical example of a model displaying superfrustration: an exponentially large degeneracy of supersymmetric ground states or, equivalently, a ground state entropy that is extensive in the system size [21–23]. By studying a two-parameter phase diagram in the vicinity of the supersymmetric point we find that it emerges as a multicritical point connecting period-4 and period-5 many-body ground states. Superfrustration arises through the collapse of a floating phase, with intermediate densities, into the supersymmetric point.

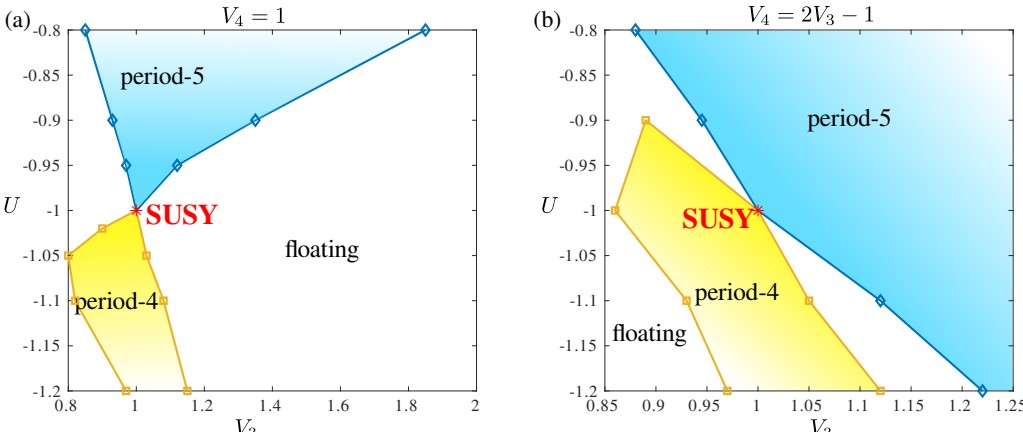

Figure 12: Phase diagram of constrained fermion model Eq. (4) on a zig-zag ladder as a function of third nearest neighbor interaction $V_3$ and on-site potential $U$ for (a) $V_4 = 1$ and (b) $V_4 = 2V_3 - 1$. The location of these cuts on a $V_3 - V_4$ plane is shown with the dash-dotted lines in Fig.10. These two phase diagrams are extracted with $N = 201$ and fixed boundary conditions with the edge sites occupied.

It would be extremely interesting to generalize our conclusion beyond the models that are supersymmetric by construction. This can be achieved, for example, by weakening the blockade. Both gapped phases as well as the floating phase are expected to survive in the soft-blockade regime and there is no immediate argument that would prevent the multicritical point to appear. It would be thus interesting to investigate how the properties such as the number of the ground states or the supersymmetric nature of this multicritical point get modified when softening the blockade.

In light of our results, it might be wise to look further into the surroundings of the super-conformal points in other systems, for instance, in chains of $SU(2)_k$ anyons [42]. In particular, it would be interesting to study what will happen to the superconformal critical point separating $\mathbb{Z}_2$ phase from the Haldane phase in the presence of perturbations that breaks translation or topological symmetry. It would also be interesting to investigate whether various $\mathbb{Z}_n$ phases can be connected via a floating phase.

The zig-zag ladder studied here is a special example of a 2D square lattice with toroidal boundary conditions (with ground state degeneracies exponential in the linear dimension of the systems). Other 2D grids, such as triangular, hexagonal or kagome, have ground state degeneracies that are exponential in the number of lattice sites. It would be most interesting to extend the analysis reported here to these 2D systems. Keeping the width of these 2D lattices sufficiently small the problem can be addressed with constrained DMRG adjusted to satisfy the blockade on a selected geometry. The nature of the ground states, however, can be addressed directly in the thermodynamic limit by means of constrained infinite projected entangled pair states [43–45]. In either case, an extensive degeneracy has to be lifted before the system can be efficiently simulated with tensor networks. The strategy proposed in this paper - to explore the vicinity of the supersymmetric point and how various phases fuse into it - seems to provide a good solution to this technical challenge.

# Acknowledgements

NC acknowledges insightful discussions with Frédéric Mila on Pokrovsky-Talapov transitions. This work has been supported by the Swiss National Science foundation. Numerical simula-

tions have been performed on the Dutch national e-infrastructure with the support of the SURF Cooperative. JM and KS acknowledge the QM&QI grant of the University of Amsterdam, supporting QuSoft.

# A    Constrained DMRG

Explicit next-nearest-neighbor blockade implies that the total dimension of the Hilbert space grows with the length of the chain as $\mathcal{H}(N) \propto 1.466^N$ [34] which is much slower than $\mathcal{H}(N) \propto 2^N$ for an unconstrained model of spin-less fermions. In order to fully profit from a restricted Hilbert space of constrained fermions, the next-nearest-neighbor blockade has been explicitly encoded into DMRG. Here we briefly recap the main set of implementation details.

First, we perform a rigorous mapping onto an effective model that spans the local Hilbert space over three consecutive sites on the original lattice as shown in Fig. 13(b). Empty circles indicate original local degrees of freedom - empty $l_1\rangle$ or occupied $l_2\rangle$. Green and blue circles corresponds to the left and right normalized MPS tensors. By spanning local degrees of freedom over three sites (dotted lines) the *local* Hilbert space increases from two states $|l_i\rangle$ sketched in Fig. 13(a) to four states $|h_i\rangle$ sketched in Fig. 13(c). States with two and more occupied sites are forbidden by the constraint. The new local Hilbert space $|h_i\rangle$ corresponds to the physical leg of the local tensor shown as a vertical solid line in Fig. 13(b). From the Fig. 13(b) it is obvious that any of the two consecutive tensors have a pair of common (or shared) sites. Three possible states of these two sites can be used as quantum labels (00), (10) and (01) for auxiliary legs that naturally create a block-diagonal structure of local tensors. The latter drastically reduces the computational cost of simulations. In the bulk, quantum labels of the left environment are changing according to the fusion graph shown in Fig. 13(d). Fusion graph can be interpreted in the following way: to the chain that ends with two empty sites on the right and therefore labeled by (00) on the right side of the chain one can add either an occupied site that would corresponds to the local state $|h_2\rangle$ and change the label of the chain to (01), or one can add another empty site that corresponds to the local Hilbert space $|h_1\rangle$ and does not change the label (loop on the top of the fusion graph); if the chain ends with an occupied site it is labeled by (01) and only an empty site with $|h_3\rangle$ can be added on its right side, in this case the label changes to (10); finally, if the chain ends with an occupied site followed by an empty site it is labeled with (10), due to two-site blockade one can add only empty site next to it that corresponds to $|h_4\rangle$ and the new label will be (00). For the right environment, i.e. when we add sites on the left side of the chain, the direction of the arrows in the fusion graph has to be reversed. An example of the label assignment on a pair of consecutive tensors is provided in Fig. 13(e).

At the next step, we have to write down the Hamiltonian Eq. (4) in terms of local matrix product operators (see e.g. Ref. [32] for the review on the MPO approach). For this one has to rewrite the spin-less fermion model given by Eq.4 in terms of the new local variables $|h_i\rangle$. For example, the local occupation number operator $n_i$ can be written in the new local Hilbert space as a $4 \times 4$ matrix $\tilde{n}_i$ with the only non-zero element $\tilde{n}_i(3,3) = 1$. The interaction terms $2V_3 n_i n_{i+3}$ and $V_4 n_i n_{i+4}$ are transformed into two- and three-body terms $2V_3 \tilde{f}_1 \tilde{f}_2$ and $V_4 \tilde{f}_1 \tilde{f}_3 \tilde{f}_2$ correspondingly, where each of the $\tilde{f}$-matrix has only one non-zero entry: $\tilde{f}_1(4,4) = 1$, $\tilde{f}_2(2,2) = 1$, and $\tilde{f}_3(1,1) = 1$.

Finally the constrained nearest- and next-nearest neighbor hopping terms transformed into $t\tilde{a}\tilde{b}\tilde{c}\tilde{d}$ + H.c. and $t\tilde{a}\tilde{o}\tilde{p}\tilde{q}\tilde{d}$ + H.c., where the only non-zero entries of the matrices are: $\tilde{a}(2,1) = 1$, $\tilde{b}(3,2) = 1$, $\tilde{c}(4,3) = 1$, $\tilde{d}(1,4) = 1$, $\tilde{o}(3,1) = 1$, $\tilde{p}(4,2) = 1$, and $\tilde{q}(1,3) = 1$.

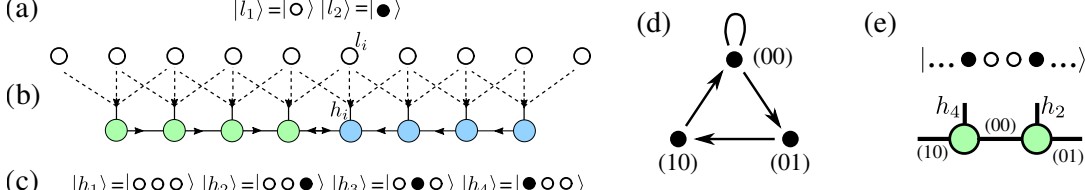

Figure 13: (a) Local Hilbert space of the original model $|l_i\rangle$. The open (filled) circle stands for an empty (occupied) site. (b) Rigorous mapping onto a model with a local Hilbert space of the MPS (blue and green tensors) spanned over three consecutive sites (empty circles) that consist of four states sketched in (c). These four states form physical bond (vertical lines) of local tensors (blue and green circles), with are contracted with each other via auxiliary bonds (horizontal lines). (d) Fusion graph for the recursive construction of the left environment (green tensors in (b)); for the right environment (blue tensors in (b)) the direction of the arrows should be inverted. (e) Example of the label assignment on two consecutive tensors written for the selected state.

With these definitions, the MPO in the bulk takes the following form:

$$
\begin{pmatrix}
\tilde{I} & . & . & . & . & . & . & . & . & . & . & . & . & . \\
\tilde{d} & . & . & . & . & . & . & . & . & . & . & . & . & . \\
\tilde{d}^{\dagger} & . & . & . & . & . & . & . & . & . & . & . & . & . \\
. & \tilde{c} & . & . & . & . & . & . & . & . & . & . & . & . \\
. & . & \tilde{c}^{\dagger} & . & . & . & . & . & . & . & . & . & . & . \\
. & . & . & t\tilde{b} & . & . & . & . & t\tilde{o} & . & . & . & . & . \\
. & . & . & . & t\tilde{b}^{\dagger} & . & . & . & . & t\tilde{o}^{\dagger} & . & . & . & . \\
. & \tilde{q} & . & . & . & . & . & . & . & . & . & . & . & . \\
. & . & \tilde{q}^{\dagger} & . & . & . & . & . & . & . & . & . & . & . \\
. & . & . & . & . & . & . & \tilde{p} & . & . & . & . & . & . \\
. & . & . & . & . & . & . & . & \tilde{p}^{\dagger} & . & . & . & . & . \\
\tilde{f}_2 & . & . & . & . & . & . & . & . & . & . & . & . & . \\
. & . & . & . & . & . & . & . & . & . & . & \tilde{f}_3 & . & . \\
-4U\tilde{n} & . & . & . & . & \tilde{a} & \tilde{a}^{\dagger} & . & . & . & . & 2V_3\tilde{f}_1 & V_4\tilde{f}_1 & \tilde{I}
\end{pmatrix},
\tag{26}
$$

where the dots mark zero entries of the tensor. Note that all entries are $4 \times 4$ matrices, so resulting MPO is a rank-4 tensor with dimensions $4 \times 4 \times 14 \times 14$. Close to the edges one has to carefully modify the MPO to properly encode the boundary terms. This requires the definition of local operators slightly different from those used in the bulk. Further details on constrained DMRG can be found in Refs. [33, 34].

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
