# Peer review of "Supersymmetry and multicriticality in a ladder of constrained fermions"

_SciPost Physics, doi:SciPost Phys. 11, 059 (2021)_

## Round 2 · Referee Report · Anonymous · 2021-6-30

Report

The authors take a starting point a fermion ladder with extensive ground-state degeneracies following from supersymmetry. They perturb the couplings, breaking the supersymmetry and most of the degeneracies, and analyze this model in great depth. Much of the phase diagram is a c=1 incommensuate phase, but the supersymmetric point is a multicritical point separating nearby period-4 and period-5 ordered phases as one might have guessed. They also find a period-3 phase where the density is at its maximum allowed value.

I think this paper is a solid piece of work, coming from an impressively thorough numerical analysis. Although there are no great surprises, it is well worth knowing how these peculiar supersymmetric points fit into a larger picture. I thus support its publication in SciPost.

I have a few comments.

-- I wouldn't say (as the authors do in the abstract) that they have "resolved the issue" of extended degeneracies in the supersymmetric models. They've certainly thoroughly understood one particular case, but I'd say the fully 2d cases are still rather mysterious. I expect the answers there to be much subtler.

-- Early on, they fix the chemical potential to be U=-1 (the supersymmetric value). I appreciated the need to make the problem tractable, but it would nice to know that nothing particularly depends on this choice. So whereas I wouldn't demand any major work, it would be nice to be reassured via a few checks that nothing dramatically changes if this fixing is relaxed a bit.

-- The authors state that in the period-4 phase there is an exact ground state along a line segment. They give the exact expression for filling exactly 1/4 (and defined a modified model governing the case where the number of sites is even but not a multiple of 4), but don't display anything specific for other cases. Is the ground state really exact in all cases, or just very close? If the former, they should explain why they haven't given the exact ground state, and if the latter, they should explain. I would expect the former, but would just like to know. Also, presumably if $U\ne -1$, they're no longer exact?

--Re the second paragraph of the conclusion: while some speculation is allowed, I worry somewhat that if the hard-core constraint is relaxed along with the supersymmetry, the multicriticality will collapse. Doesn't Landau theory suggest that? If it doesn't, it would be really interesting to understand why not. Probably the authors should clarify this (or correct me if I've misunderstood).

-- Re the third paragraph of the conclusion: At least as I understand it, the results of Trebst et al are protected by the "topological symmetry" imposed. So whereas another look at these results would be an excellent idea, I wouldn't expect them to change without breaking the symmetry.

-- Re the fourth paragraph of the conclusion: by "problems" what do they mean?? Extensive ground-state degeneracy and what else?

---

## Round 2 · Referee Report · Anonymous · 2021-7-4

Strengths

1-systematic study of the phases in a spin-less fermion model with a supersymmetric multi-critical point
2-clear phenomenological picture of the ground and excited states in limiting cases
3-criticalities in floating phases

Weaknesses

1-The explanation for the algorithm in Appendix A can be improved.

Report

The authors study the phase diagram of a constrained spinless fermion model on a zigzag ladder. The model possesses a supersymmetric point at which the ground state degeneracy is extensive. The perturbation away from the supersymmetric point lifts this massive degeneracy and leads to either gapless or gapped phase, depending on the relative strength of the third and fourth neighbor interactions. An extensive DMRG study reveals that (i) the gapless phase is an incommensurate Luttinger liquid phase described by c=1 CFT and (ii) the transition between the gapless and gapped phases is in the Pokrovsky-Talapov universality class. In addition, in some limiting cases, the authors give clear phenomenological pictures of the ground states and excited states, which are consistent with what they found numerically. I think the results are interesting, and the paper is written very nicely. Thus, I recommend the manuscript for publication after the authors consider the following comments and suggestions.

- PXP model
In the first paragraph of the Introduction, the authors classified the PXP model as a model with a kinetically constrained hopping. However, the Hamiltonian of the PXP model consists solely of the constrained local spin-flip terms and does not have any kinetic term. Thus, I think the corresponding sentence should be rephrased with more careful wording.

- Eq. (7)
It is not easy to understand what this equation means unless one writes down the 1-particle Hamiltonian. As far as I understand, N=5 is special in that the 1-particle Hamiltonian is written as a 5x5 matrix with all matrix elements 1. I suggest that the authors write down the explicit Hamiltonian just like Eq. (9).

- Eq. (9)
There must be a typo in this equation. I think the (3,3) element of this matrix should be 1. Otherwise, {\bm v}_{\rm GS} below Eq. (9) cannot be a zero-energy state.

- Criticality at the supersymmetric point
I wonder if the authors can study the nature of the criticality right at the supersymmetric point. I know the massive ground state degeneracy at the point makes the standard stuff like the central charge meaningless. Nevertheless, I still wonder if one can talk about the criticality of the ground state sector by sector. Are there any particular fermion densities N_f/N at which the low-energy states are not so degenerate, and the standard CFT machinery applies? (This question might be related to the second comment raised by the other referee: are there any chemical potentials for which the standard analysis makes sense?)

- Boundary of the period-3 phase
I like the phenomenological argument presented in Sec. 3.3, as it is intuitive and easy to understand. However, I do not quite see how the phase boundaries obtained by that argument are consistent with the actual boundaries obtained numerically. In the current manuscript, the authors just state that they agree with each other. I wonder if the authors can show an enlargement of the phase diagram around the phase boundary of the period-3 phase and show the comparison between numerical and phenomenological results, which would be helpful.

- Uniqueness of the ground state(s)
The authors found that the states Eq. (18) are (19) ground states of the model along the line V_3 = V_4 in the period-4 phase. I am curious about whether they are the unique ground states or there are some other ground states. Can the authors prove that there are no other ground states by using the Perron-Frobenius theorem etc.?

- Appendix A
This appendix is just hard to understand. I have almost no idea about what Figures 12 (a)-(e) mean. What does the fusion graph mean? Does the MPO here mean the MPO expression for the Hamiltonian? I would like to suggest that the authors elaborate on them in both the main text and the caption of Fig. 12.

Requested changes

1. First paragraph of Sec. 3.1:
know in the literature ... -> known in the literature ...

---

## Round 3 · Referee Report · Anonymous (Referee 2) · 2021-8-7

Report

I have gone through the revised manuscript and the authors' response to the the referees. I think the authors have addressed all the comments raised by the referees and improved the quality/readability of the paper considerably. Thus, I think the current manuscript is almost ready for acceptance. I only have a minor comments which could be addressed before the publication.

1. Reply to my first comment in the previous report
In their response to my comment #1 (regarding the PXP model), the authors claim that they have modified the first paragraph of the Introduction appropriately. However, as far as I see, the paragraph is exactly identical to that of the previous submission.

Requested changes

See the comment in Report.

---

## Round 3 · Referee Report · Anonymous (Referee 1) · 2021-8-30

Report

The authors have addressed appropriately the issues raised in both referees' reports. It thus should be published in SciPost.

Requested changes

there's a typo in the next-to-last paragraph, it should read "translation or topological symmetry", not "of".

---

## Round 3 · Author Response

Dear Editors,

please find attached the updated version of our manuscript “Supersymmetry and multicriticality in a ladder of constrained fermions”. We would like to thank the Referees for their stimulating questions and constructive criticism and are pleased to note that both support a publication. Below we address in detail all the Referees' concerns and we hope that with these changes made, our manuscript can now be accepted for publication in SciPost.

Sincerely yours,

the authors

Remark: The numbering refers to the new version of the manuscript.

Referee 1:

-- I wouldn't say (as the authors do in the abstract) that they have "resolved the issue" of extended degeneracies in the supersymmetric models. They've certainly thoroughly understood one particular case, but I'd say the fully 2d cases are still rather mysterious. I expect the answers there to be much subtler.

We have changed “resolve” to a more appropriate “address” in the abstract.

-- Early on, they fix the chemical potential to be U=-1 (the supersymmetric value). I appreciated the need to make the problem tractable, but it would nice to know that nothing particularly depends on this choice. So whereas I wouldn't demand any major work, it would be nice to be reassured via a few checks that nothing dramatically changes if this fixing is relaxed a bit.

We have extended our analysis along the lines suggested by the Referee as we now describe in a new subsection 3.6 and show in Fig. 12. As we discuss in the text, the nature of the supersymmetric point remains preserved also for deviations from the exact supersymmetry U \neq 1 in that the period-4 and period-5 phase remain always separated by a floating phase. Specifically, we have performed a study along two cuts in the V_4 – V_3 plane indicated by the dash-dotted lines in Fig. 10, i.e. one performing a cut crossing the supersymmetric point from a floating to a floating phase and the other from period-4 to period-5 phase with the corresponding phase diagram in the U-V_3 plane shown in Fig. 12a and Fig. 12b, respectively.

-- The authors state that in the period-4 phase there is an exact ground state along a line segment. They give the exact expression for filling exactly 1/4 (and defined a modified model governing the case where the number of sites is even but not a multiple of 4), but don't display anything specific for other cases. Is the ground state really exact in all cases, or just very close? If the former, they should explain why they haven't given the exact ground state, and if the latter, they should explain. I would expect the former, but would just like to know. Also, presumably if U≠−1 , they're no longer exact?

As the Referee points out, the nature of the exact ground state along the V_3=V_4 line within the period-4 phase is a subtle issue depending on the boundary conditions and the specific length of the ladder with four distinct possibilities, namely N mod 4 = 0,1,2,3. What they have in common is that the state of the type Eq. (18) is no more an eigenstate of H but picks up corrections near the edges. As we now explain in Sec. 3.4, the effect of the boundaries on the nature of the ground state can be understood in two complementary ways – either one has to deform the bulk Hamiltonian H at the boundaries, which we demonstrate on a specific example of N = 4 N_f – 2, cf. Eq. (20), to make the state (18) an exact eigenstate. Or, alternatively, one has to modify the state (18) itself to make it an eigenstate of H. Since the effect of the open boundaries is typically a modification of the particle densities near the edges, rather than searching for an analytical expression for such a ground state, we provide a numerical example in Eq. (21) to demonstrate this point. In addition, for odd N the VBS states require a breaking of the left-right symmetry, which we indeed observe in the output of our numerical DMRG procedure.

--Re the second paragraph of the conclusion: while some speculation is allowed, I worry somewhat that if the hard-core constraint is relaxed along with the supersymmetry, the multicriticality will collapse. Doesn't Landau theory suggest that? If it doesn't, it would be really interesting to understand why not. Probably the authors should clarify this (or correct me if I've misunderstood).

The supersymmetric point in our model appears as the multicritical point between the gapless floating phase and two gapped phases. Weakening the blockade does not close the gap immediately, so we expect these two phases to survive, at least for a sufficiently strong repulsion between the nearest neighbors. It is also natural to expect that weakening the blockade simplifies quantum fluctuations, so the floating phase separating period-4 and period-5 phases will remain. As we wrote in the manuscript, it is not clear under which conditions the multicriticality will be lifted, but it is possible that in the 7-dimensional parameter space of the model exists a line or even a surface along which the multicriticality is preserved by the fine-tuning of the parameters.

-- Re the third paragraph of the conclusion: At least as I understand it, the results of Trebst et al are protected by the "topological symmetry" imposed. So whereas another look at these results would be an excellent idea, I wouldn't expect them to change without breaking the symmetry.

The referee is right. We have modified the manuscript accordingly.

-- Re the fourth paragraph of the conclusion: by "problems" what do they mean?? Extensive ground-state degeneracy and what else?

We have modified the wording to emphasize that we refer to the possible numerical approaches to address the nature of the highly degenerate ground states appearing in the supersymmetric lattice models in higher dimensions.

Referee 2:

-- PXP model
In the first paragraph of the Introduction, the authors classified the PXP model as a model with a kinetically constrained hopping. However, the Hamiltonian of the PXP model consists solely of the constrained local spin-flip terms and does not have any kinetic term. Thus, I think the corresponding sentence should be rephrased with more careful wording.

We have modified the corresponding sentence to remove the inaccuracy pointed out by the Referee.

-- Eq. (7)
It is not easy to understand what this equation means unless one writes down the 1-particle Hamiltonian. As far as I understand, N=5 is special in that the 1-particle Hamiltonian is written as a 5x5 matrix with all matrix elements 1. I suggest that the authors write down the explicit Hamiltonian just like Eq. (9).

We have now extended the discussion of this point and have now included explicitly the Hamiltonian for periodic boundary conditions, cf. Eq. (7).

-- Eq. (9)
There must be a typo in this equation. I think the (3,3) element of this matrix should be 1. Otherwise, {\bm v}_{\rm GS} below Eq. (9) cannot be a zero-energy state.

Indeed, we thank the Referee for spotting this typo which we have now corrected, cf. Eq. (9) in the new version of the manuscript.

-- Criticality at the supersymmetric point
I wonder if the authors can study the nature of the criticality right at the supersymmetric point. I know the massive ground state degeneracy at the point makes the standard stuff like the central charge meaningless. Nevertheless, I still wonder if one can talk about the criticality of the ground state sector by sector. Are there any particular fermion densities N_f/N at which the low-energy states are not so degenerate, and the standard CFT machinery applies? (This question might be related to the second comment raised by the other referee: are there any chemical potentials for which the standard analysis makes sense?)

We have added a paragraph addressing this issue at the end of section 3.2.

-- Boundary of the period-3 phase
I like the phenomenological argument presented in Sec. 3.3, as it is intuitive and easy to understand. However, I do not quite see how the phase boundaries obtained by that argument are consistent with the actual boundaries obtained numerically. In the current manuscript, the authors just state that they agree with each other. I wonder if the authors can show an enlargement of the phase diagram around the phase boundary of the period-3 phase and show the comparison between numerical and phenomenological results, which would be helpful.

We have included the phase boundaries as estimated by the single and double defect instabilities discussed in Sec. 3.3 as dashed green lines in the phase diagram Fig. 2.

-- Uniqueness of the ground state(s)
The authors found that the states Eq. (18) are (19) ground states of the model along the line V_3 = V_4 in the period-4 phase. I am curious about whether they are the unique ground states or there are some other ground states. Can the authors prove that there are no other ground states by using the Perron-Frobenius theorem etc.?

For OBC, V_3=V_4=a (with a close to but smaller than 1) and N even, H has a unique GS, which takes the form of a symmetrically placed VBS pattern with corrections near the edges. (We lack a formal proof but observe that results for small systems are very stable in their dependence on N due to the small correlation length in the VBS states.) For N odd the VBS patterns require a breaking of the left-right symmetry and are thus never exact GS in finite size. We have modified our text to stress these points, cf. also our answer to the similar question (point #3) raised by the Referee 1 above.

-- Appendix A
This appendix is just hard to understand. I have almost no idea about what Figures 12 (a)-(e) mean. What does the fusion graph mean? Does the MPO here mean the MPO expression for the Hamiltonian? I would like to suggest that the authors elaborate on them in both the main text and the caption of Fig. 12.

Fusion graph is a schematic representation of the sequence according to which the quantum labels of auxiliary legs of the local tensors change in the constrained DMRG upon adding one particle to an environment tensor.
We elaborated on the interpretation of the fusion graph and added some examples on how it can be used to recurrently construct the environment tensors. We also clarified the caption of Fig.12 (now Fig.13). MPO is indeed an MPO expression of the Hamiltonian, that was re-written in terms of the new local Hilbert space |h_i>. We clarified this in the Appendix.

-- First paragraph of Sec. 3.1: know in the literature ... -> known in the literature ...

We thank the Referee for spotting the typo which we have now fixed.

---

## Editorial Decision

published